# Relation Extraction for Constructing Knowledge Graphs: Enhancing the Searchability of Community-Generated Digital Content (CGDC) Collections

Martin Marinov[1], Youcef Benkhedda[1], Ewan Hannaford[2], Marc Alexander[2], Goran Nenadic[1] and Riza Batista-Navarro[1,*]

[1]*Department of Computer Science, University of Manchester, Oxford Road, Manchester M13 9PL, UK*
[1]*School of Critical Studies, University of Glasgow, Glasgow, G11 6EW, UK*

## Abstract

Much of people's understanding of their cultural heritage is facilitated by the curation and preservation of community-generated digital content (CGDC): archival collections that were created for, with and by local communities. However, communities employ their own conventions in storing and publishing their content. Given this and the fact that semantic information tends to be buried within textual descriptions, CGDC archives are currently siloed and obscured, thus making it difficult for end-users (e.g., members of the public and researchers) to search for fine-grained information (e.g., "Where did Alfred Edward Julian work?"). In this paper, we propose to represent the information within CGDC archives in the form of knowledge graphs. To enable the construction of such knowledge graphs at scale, we developed a zero-shot approach for relation extraction, which we cast as a natural language inference (NLI) problem. Specifically, for each of the 20 relation types drawn from Wikidata that we have identified as relevant to CGDC, we created a premise-hypothesis pair that is presented to an NLI model that determines whether entailment (and thus the relation type) holds. The premise is a sentence from the natural language description and the hypothesis is automatically generated using a template based on each of the relation types. We present the results of comparing and combining three different transformer-based models that were already fine-tuned for the NLI task, namely, DeBERTa, BART and T5.

## Keywords

Relation Extraction, Zero-shot Prompting, Transformer Models, Knowledge Graphs, Cultural Heritage

## 1. Introduction

Cultural heritage is preserved and passed on to future generations through archival collections and their digitalisation. While enormous efforts (e.g., The National Archives of the UK, Europeana) have been put into making such collections available to the public, a significant part of people's cultural heritage is represented only in community-generated digital content (CGDC): digital-born archive collections developed for, with and by communities [1]. For example, in the UK, around 5000 community history projects have been funded by the National Lottery Heritage Fund (NLHF), allowing many communities to explore and preserve their history and heritage, leading to the proliferation of CGDC collections.

These communities tend to follow their own conventions to store and represent their CGDC collections. As a result, most of the rich semantic information contained within these collections remain buried within textual metadata, such as the titles and descriptions of CGDC items. The description of a photograph of a local landmark, for instance, might mention the person or organisation who was responsible for building that landmark—information that is potentially of interest to researchers or members of the public and yet obscured within text. In this work, we propose to transform the information within the textual metadata of CGDC collections into a knowledge graph, in order to make CGDC more searchable and queryable. To this end, we investigated zero-shot approaches to relation extraction (RE) as a means for automatically curating such knowledge graphs, thus eliminating the

*DL4KG'24: Deep Learning and Large Language Models for Knowledge Graphs, August 26, 2024, Barcelona, Spain*
*Corresponding author.

✉ m.marinov0101@gmail.com (M. Marinov); youcef.benkhedda@manchester.ac.uk (Y. Benkhedda);
ewan.hannaford@glasgow.ac.uk (E. Hannaford); marc.alexander@glasgow.ac.uk (M. Alexander);
gnenadic@manchester.ac.uk (G. Nenadic); riza.batista@manchester.ac.uk (R. Batista-Navarro)

need for any training data. Specifically, we cast RE as a natural language inference (NLI) task, allowing us to take advantage of three pre-trained transformer-based language models: DeBERTa [2], BART [3] and T5 [4]. Although RE has been addressed using zero-shot approaches in other domains [5], to the best of our knowledge, the use of such has been under-explored in the cultural heritage domain. Cetoli [6] and Chen and Li [7] proposed zero-shot RE approaches for the general domain. Meanwhile, Tang et al. [8] investigated various pre-trained transformer-based language models for RE in ancient Chinese history documents; however, their models required some form of training (either by fine-tuning or chain-of-thought prompting). Most similar to our work is that of Tan et al. [9], who also constructed knowledge graphs based on entity relations extracted from Chinese cultural heritage texts. However, their work is rule-based and did not explore the use of transformer-based models.

## 2. Relation-Annotated Dataset

To support our experimentation with transformer-based RE models, we constructed a dataset of CGDC textual metadata where relationships between named entities were manually annotated.

### 2.1. Relation Types

The work of Benkhedda et al. [10] considered the following five entity types as relevant to CGDC: `Person`, `Organisation`, `Location`, `Miscellaneous` and `Date`. Drawing inspiration from their work, we decided to focus on relation types involving any of these entity types. For this purpose, instead of creating our own relation types, we leveraged Wikidata properties [11]. Initially, a total of 29 Wikidata properties (relation types) were identified as relevant to CGDC. Our annotation process (described in Section 2.2), however, revealed that nine of these types were very rarely encountered in our dataset. Thus, in the end, after keeping only the types with at least three labelled examples, 20 relation types were retained. We refer the reader to Appendix A for a full listing of these relation types.

### 2.2. Data Collection and Annotation

In our work, we utilised the CGDC dataset that was annotated with named entities (and their corresponding identifiers in Wikidata) constructed by Benkhedda et al. [10]. The subset of their dataset that was made publicly available consists of 100 documents, where each document corresponds to the concatenation of the title and description of a CGDC item. Half of the dataset (50 documents) came from the Morrab Library Photographic Archive comprised of thousands of digitised photographs that capture Cornish culture and history.[1] The other 50 documents were drawn from People's Collection Wales (PCW),[2] an online platform for gathering various media (e.g., photographs, documents, audio and video recordings) from individuals and community groups who wish to contribute items related to Welsh culture and history.

We then designed a simple relation annotation scheme, where we defined a relation as consisting of a *head* and *tail* entity, which are linked according to a relation type. For each relation type, our scheme specifies the possible entity types for each of the head and tail entities. For example, for the `notable_work` relation type, there are three possible head-tail entity combinations: `Person-Miscellaneous` (applies when a person created a miscellaneous entity such as a work of art), `Organisation-Miscellaneous` (if the creator is an organisation) and `Person-Location` (applies when a person is known for building or designing a location such as a church). Appendix A provides all the valid head-tail entity type combinations for each CGDC relation type.

The brat tool [12] was configured to enable the manual annotation of our relation types of interest, with the entity type constraints for each relation type specified. Two annotators (the second and last authors of this paper) independently labelled entity relations in all of the 100 documents. The inter-annotator agreement (IAA) between the two annotators was determined to be 0.48 in terms of

---

[1]https://photoarchive.morrablibrary.org.uk/
[2]https://www.peoplescollection.wales/

Cohen's Kappa [13], which is considered to be substantial agreement [14], considering that there are 20 possible classes. The annotated set of 100 documents was then expanded by labelling the named entities and relations in 137 further documents drawn from the PCW collection. Similar to the first set of documents, two annotators (the third and last authors of this paper) independently annotated the entity relations in all of the 137 documents and a similar level of IAA (i,e., 0.49 in terms of Cohen's Kappa) was obtained. In our experiments for evaluating the performance of transformer-based RE models, we utilise all the 237 relation-labelled CGDC documents.

## 3. Methods

The RE task can be defined as follows: given a sentence $s$ and a pair of entities $e1$ and $e2$ contained within that sentence, as well as a set of classes $C$, an RE model should identify one class $c \in C$ that best describes the relationship between $e1$ and $e2$ based on $s$. In this work, we are concerned with 20 relation types that are relevant to CGDC. However, apart from these, we also include an additional class that we refer to as None Of The Above or NOTA, to which pairs of entities that are not related according to any of our 20 CGDC types belong. Thus, there are a total of 21 classes in $C$.

### 3.1. Models

A number of transformer-based models that were already fine-tuned for the NLI task were employed to extract relations between entities in a zero-shot manner. Each of these models is described below.

**DeBERTa** (which stands for Decoding-Enhanced BERT with disentangled Attention) improves upon the encoder-only BERT model by employing a disentangled attention mechanism [15]. In this work, we employed version 3 of the DeBERTa model [2] that was fine-tuned on NLI datasets.[3] This particular model can be employed in a zero-shot classification manner, whereby the model classifies an input sequence according to a given set of classes (labels), also providing a probability for each class.

**BART** is an encoder-decoder model that combines a BERT-like encoder and an auto-regressive decoder similar to GPT. It has demonstrated satisfactory performance on both text generation and comprehension tasks (such as text classification) [3]. We employed a version of the BART model that was fine-tuned for NLI.[4] Similar to our chosen DeBERTa model, this model can be employed in a zero-shot classification manner and provides probabilities together with predicted labels.

**T5** stands for Text-to-Text Transfer Transformer [4], which casts many downstream NLP tasks (including NLI) as a sequence-to-sequence modelling problem, following an encoder-decoder architecture. An extra extra large (XXL) version of T5 that was fine-tuned for the NLI task was employed in our experiments [16].[5] Unlike the DeBERTa and BART models, this model does not provide any probability values together with its predicted labels.

Importantly, we investigated an ensemble model, henceforth referred to as **Ensemble**, that is a combination of the above three models. The prediction of this ensemble model was determined by taking the majority vote amongst the constituent transformer-based models.

### 3.2. Experimental Setup

As mentioned above, we cast RE as an NLI task, whereby two sentences are provided to a model as input: a *premise* and a *hypothesis*. If based on the premise, the hypothesis is true, then the model should detect that an *entailment* relation holds between the two sentences. Otherwise, there is *no entailment* relation between the two sentences. To frame RE as an NLI task, an input sentence $s$ is considered to be the premise, while a hypothesis is automatically generated by populating a sentence template—that is predefined for a particular relation type—with input entities $e1$ and $e2$. The premise-hypothesis pair is then presented to an NLI model. If the model detects entailment, then we say that the relation type

---

[3]https://huggingface.co/cross-encoder/nli-deberta-v3-large
[4]https://huggingface.co/facebook/bart-large-mnli
[5]https://huggingface.co/google/t5_xxl_true_nli_mixture

**Table 1**
Examples of premise-hypothesis pairs used as inputs to the NLI models.

| Relation Type | Template | Premise (Input Sentence with Entities Underlined) | Template-generated Hypothesis |
|---|---|---|---|
| inception | `<Org\|Loc\|Misc>` was created on `<Date>` | *Operating from 1923, the Lady Magdalen was just one of the ferries that would carry passengers and cars across the Cleddau River.* | *Lady Magdalen was created on 1923.* |
| operating_area | `<Misc>` operated in `<Loc>` | *Operating from 1923, the Lady Magdalen was just one of the ferries that would carry passengers and cars across the Cleddau River.* | *Lady Magdalen operated in Cleddau River.* |

(corresponding to the template) holds between the input entities. In Table 1, we provide examples of premise-hypothesis pairs, in which the hypothesis was generated based on a template. The complete set of templates is provided as part of our codebase.[6]

In preparation for applying the NLI models in a zero-shot manner, we firstly segment each document (in the evaluation dataset) into individual sentences. For each of these input sentences, all possible pairs of entities (contained within a sentence) are created. For every entity pair, a hypothesis is generated for each relation type. Each of these generated hypotheses is then paired up with the premise (the input sentence). Finally, our NLI models take every premise-hypothesis pair as an input sample that is then classified as being characterised by entailment or not. If entailment is not detected (with a probability of at least 0.40 in the case of the BART and DeBERTa models) for any of the relation types, we assign the NOTA label to the input sample. In cases where entailment is detected by the model for more than one class (relation type), we simply take the class with the highest probability value, as provided by the BART and DeBERTa models. As the T5 model does not output any probability values, we implemented post-processing rules that specify which relation types should take precedence in case of ties.

## 4. Evaluation and Error Analysis

In this section, we report the results of applying our chosen NLI models to the RE task in the zero-shot manner described in the preceding section. As previously mentioned, entity pairs were exhaustively created based on our evaluation data (the 237 CGDC documents) to form the input samples. This resulted in 5938 entity pairs that the NLI models need to classify according to relation type (or the lack thereof, in which case the NOTA class applies). Out of those, 5435 (or 92%) belong to the NOTA class.

In Table 2, we report the performance obtained by each of our models in terms of the standard metrics of precision, recall and F1-score. To avoid skewing the results towards the over-represented NOTA class, we report the performance separately for the 20 CGDC relation types, and the NOTA class. Weighted macro-averaging was employed in reporting combined performance for the 20 CGDC relation types.

With respect to the 20 CGDC relation types, the Ensemble model obtained the best weighted macro-averaged F1-score of 0.487. Considering the IAA of 0.48-0.49 that we obtained (see Section 2.2), the model comes close to upper-bound performance. Meanwhile, the models obtained F1-scores as high as 0.914 (for T5) on the NOTA class. This is particularly impressive considering that it is well-known that handling NOTA cases is a challenging task [17].

We manually examined some cases where the models made incorrect predictions. For instance, in the sentence *"She was born on 10th March 1833 at Market Rasen, Lincolnshire, the eldest daughter of Henry Albert Browne and Frances Margaret Nicholson"*, no relation (NOTA) holds between the entities *"Frances Margaret Nicholson"* and *"Henry Albert Browne"*, according to the gold standard annotations. Both T5

---

[6]Our code and annotations (for the 237 in CGDC documents in our dataset) are available at https://github.com/OurHeritageOurStories/cgdc_re.

**Table 2**

Performance of the RE models in terms of weighted macro-averaged scores. For the CGDC types, the average was taken over all 20 relation types while for the `None Of The Above (NOTA)` label, the scores simply correspond to the performance on this one label.

|  | Model | Precision | Recall | F1-score |
|---|---|---|---|---|
| CGDC Relation Types | BART | 0.293 | 0.817 | 0.403 |
|  | DeBERTa | 0.420 | 0.718 | 0.454 |
|  | T5 | 0.474 | 0.620 | 0.472 |
|  | Ensemble | 0.431 | 0.779 | **0.487** |
| NOTA | BART | 0.993 | 0.483 | 0.650 |
|  | DeBERTa | 0.990 | 0.638 | 0.776 |
|  | T5 | 0.979 | 0.857 | **0.914** |
|  | Ensemble | 0.991 | 0.665 | 0.796 |

and BART predicted the `spouse` label for this entity pair (whereas DeBERTa successfully predicted NOTA). One can, however, argue that even the predictions by T5 and BART are not necessarily incorrect. The interpretability of such results are rather subjective and can be considered as either correct or incorrect based on people's opinion or the intended use of the models. Overall, it was observed that the models tended to detect or infer implicit relations that human annotators might otherwise miss.

## 5. Knowledge Graph Curation

Taking the predictions of the Ensemble model over the 237 documents in our dataset, we populated a knowledge graph (KG) whereby vertices represent named entities and edges represent any relations detected between them. Here, only entities that were manually assigned Wikidata IDs (as part of gold standard entity linking annotations) were included, to ensure that entities in the resulting KG are normalised.

To create the knowledge graph, we utilised the Neo4j framework[7], a graph database management tool. Using the Cypher query language, we were able to query the resulting knowledge graph. Figure 1 shows a visualisation of the results of an example query for a use case where the relationships of a particular person of interest (King Edward) with other entities have been retrieved.

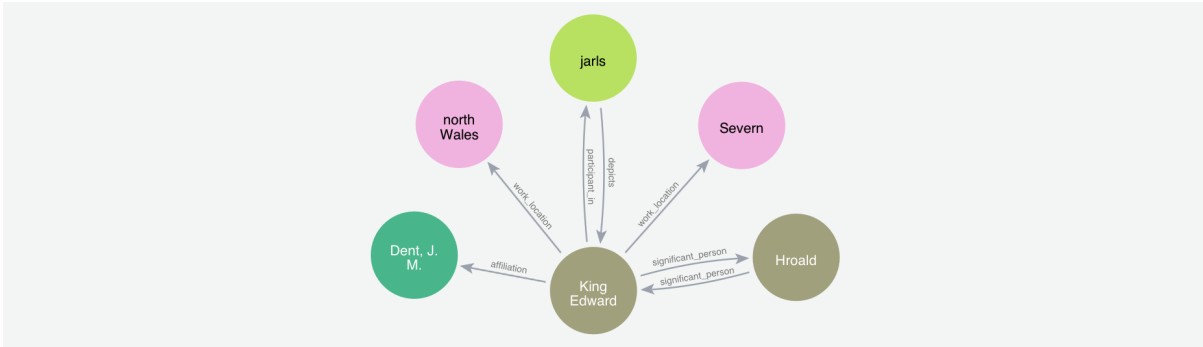

**Figure 1:** Results obtained by querying the populated knowledge graph for information on entities related to King Edward.

## 6. Conclusions

In this paper, we demonstrate how a zero-shot approach based on NLI can be employed to extract entity relations in CGDC textual metadata. Our findings show that, based on evaluation using a dataset of

---

[7]https://neo4j.com/

237 CGDC documents, an ensemble of three different transformer-based models (BART, DeBERTa and T5) obtains the best weighted macro-averaged F1-score for 20 CGDC relation types. The knowledge graph that was constructed based on automatically extracted relations provides a means for searching for information that is otherwise buried within CGDC textual metadata. Our future work will focus on expanding the CGDC dataset to include more relation-annotated documents that will allow for more robust evaluation, including experimentation and comparison with closed-sourced large language models.

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

## A. Details of CGDC Relations

The relation types relevant to CGDC, their equivalent human-readable label, their Wikidata identifiers (linked to a page with their definitions and synonyms) and the types of head and tail entities involved.

| Wikidata Property/ Relation Type | Human-readable Label | Wikidata ID | Head Entity Type | Tail Entity Type |
|---|---|---|---|---|
| affiliation | connected to | P1416 | Person | Organisation |
| | | | Organisation | Organisation |
| date_of_birth | born on | P569 | Person | Date |
| date_of_death | died on | P570 | Person | Date |
| depicts | shows | P180 | Miscellaneous | Person |
| | | | Miscellaneous | Location |
| | | | Miscellaneous | Organisation |
| | | | Miscellaneous | Miscellaneous |
| employer | worked for | P108 | Person | Person |
| | | | Person | Organisation |
| inception | began to exist on | P571 | Organisation | Date |
| | | | Location | Date |
| | | | Miscellaneous | Date |
| location | happened in | P276 | Miscellaneous | Location |
| located_in | located in | P706 | Location | Location |
| member_of | member of | P463 | Person | Organisation |
| notable_work | created or built | P800 | Person | Miscellaneous |
| | | | Organisation | Miscellaneous |
| | | | Person | Location |
| occupation | worked as | P106 | Person | Miscellaneous |
| operating_area | operated in | P2541 | Miscellaneous | Location |
| participant_in | participated in | P1344 | Person | Miscellaneous |
| | | | Organisation | Miscellaneous |
| partnership_with | collaborated with | P2652 | Organisation | Organisation |
| place_of_birth | born in | P19 | Person | Location |
| point_in_time | happened on or as of | P585 | Miscellaneous | Date |
| | | | Location | Date |
| residence | lived in | P551 | Person | Location |
| significant_person | wrote to, spoke to or met | P3342 | Person | Person |
| spouse | married to | P26 | Person | Person |
| work_location | worked in | P937 | Person | Location |
| | | | Organisation | Location |