# OpenReview forum: "Relation Extraction for Constructing Knowledge Graphs: Enhancing the Searchability of Community-Generated Digital Content (CGDC) Collections"
_KDD.org/2024/Workshop/DL4KG — DL4KG 2024_

### Official Review · Reviewer_xArL · 2024-07-03
**The authors propose a reasonable approach to use existing techniques for creating KGs from unstructured data within the domain of CGDC Collections.**

**Rating:** 7
**Confidence:** 4

**Review:**

The authors describe a method to construct KGs via Relation Extraction (RE). Specifically, they propose performing RE as a Natural Language Inference (NLI) Task. For every relation type, the authors construct a template which they use to construct Hypotheses for the NLI task. The description of CGDC elements are used as Premises. By performing the NLI task with this setup, the correct relation type for the entities in the CGDC descriptions are extracted. This setup has the advantage that it can be performed with fine-tuned NLI models, without requiring extra training.
The presented approach is a combination of established techniques, but still provide a sufficient amount of novelty. The presented techniques show potential for creating KGs from unstructured sources in general. The approach is clearly described and the presentation makes it easy to follow the line of arguments.
Cons:
-	Limited evaluation (no comparison to different approaches or datasets)
Pros:
-	Showcasing the application of established techniques on new domains
-	Clear presentation

---

### Official Review · Reviewer_j8FM · 2024-07-07
**Review paper Relation Extraction for Constructing Knowledge Graphs: Enhancing the Searchability of Community-Generated Digital Content (CGDC) Collections**

**Rating:** 5
**Confidence:** 4

**Review:**

This (short) paper presents a zero-shot approach for extracting relations from archives, as a basis for the construction of a knowledge graph (very shortly described in the paper). The aim is to make such archives more searchable and queryable. 20 types of relations from Wikidata have been considered in experiments which have been run on three different transformer-based methods (DeBERTa, BART and T5) along with their combination.

The literature on knowledge extraction in cultural heritage is huge and the authors claim that the use of a zero-shot approach has been fully exploited in the domain. The proposed approach is however quite classical (template-based). The results are not concluding and the authors do not go further on their discussion and future developments. Some details are also missing: how the relevantance of the 20 types of the CGDC has been established? how to improve inter-annotator agreement? how the sentence templates has been constructed?

---

### Official Review · Reviewer_Xxo9 · 2024-07-08

**Rating:** 8
**Confidence:** 4

**Review:**

The paper proposes a relation extraction approach based on zero shot learning. It uses masked language models like BART DeBERTA, T5. The paper targets very interesting problem of knowledge graph construction with an application towards digital humanities which is already under explored.

- It would be good to justify the choice of language models.
- Did authors try out large language models: https://aclanthology.org/2023.acl-long.868.pdf
- The inter-annotator agreement is a bit low 0.48

---

### Decision · Program_Chairs · 2024-07-09

Accept